# Plant-frugivore network simplification under habitat fragmentation leaves a small core of interacting generalists

Wande Li [1], Chen Zhu[2], Ingo Grass [3], Diego P. Vázquez [4,5], Duorun Wang[1], Yuhao Zhao[1], Di Zeng [1], Yi Kang[1], Ping Ding [2] & Xingfeng Si [1✉]

Habitat fragmentation impacts seed dispersal processes that are important in maintaining biodiversity and ecosystem functioning. However, it is still unclear how habitat fragmentation affects frugivorous interactions due to the lack of high-quality data on plant-frugivore networks. Here we recorded 10,117 plant-frugivore interactions from 22 reservoir islands and six nearby mainland sites using the technology of arboreal camera trapping to assess the effects of island area and isolation on the diversity, structure, and stability of plant-frugivore networks. We found that network simplification under habitat fragmentation reduces the number of interactions involving specialized species and large-bodied frugivores. Small islands had more connected, less modular, and more nested networks that consisted mainly of small-bodied birds and abundant plants, as well as showed evidence of interaction release (i.e., dietary expansion of frugivores). Our results reveal the importance of preserving large forest remnants to support plant-frugivore interaction diversity and forest functionality.

[1] Institute of Eco-Chongming (IEC), Zhejiang Tiantong Forest Ecosystem National Observation and Research Station, School of Ecological and Environmental Sciences, East China Normal University, 200241 Shanghai, China. [2] MOE Key Laboratory of Biosystems Homeostasis and Protection, College of Life Sciences, Zhejiang University, 310058 Hangzhou, Zhejiang, China. [3] Ecology of Tropical Agricultural Systems, University of Hohenheim, 70599 Stuttgart, Germany. [4] Argentine Institute for Dryland Research, CONICET & National University of Cuyo, 5500 Mendoza, Argentina. [5] Faculty of Exact and Natural Sciences, National University of Cuyo, M5502JMA Mendoza, Argentina. ✉email: sixf@des.ecnu.edu.cn

The interactions between plants and their frugivores (i.e., potential seed dispersers) play an essential role in the maintenance of biodiversity and ecosystem functioning[1–3]. Habitat fragmentation reduces habitat size, decreases landscape connectivity and increases habitat isolation[4], and thereby may represent a major driver of the loss of species and interaction diversity[2,5,6]. For example, habitat fragmentation may hamper plant-frugivore interactions, triggering the loss of seed dispersal services[2,7,8]. However, empirical studies examining the effects of habitat fragmentation at a community level are still limited, mainly due to the lack of high-quality data[9]. This is especially the case for studies focusing on the complex interaction networks emerging from plants and frugivorous animals that potentially act as their seed dispersers[10,11]. A better understanding of how habitat fragmentation influences plant-frugivore networks (PFNs) could also help policymakers fulfil conservation strategies and guide restoration efforts[12,13].

Environmental changes may influence species interaction networks[14,15], which in turn may affect community stability and their resistance to species loss[16–18]. Robustness refers to the tolerance of communities to species extinctions[19]. Previous studies have shown that certain network properties (e.g., connectance, modularity and nestedness) may impact network robustness[17,18,20]. Connectance quantifies the proportion of observed links to potential links in a network[21] and can enhance network robustness via increased redundancy of interaction partners[19] (but see Vieira et al.[22]). Modularity refers to the tendency of a network to comprise subgroups of closely interacting species[23]. The number of interactions within modules tends to be higher than between modules. Theoretically, higher modularity may buffer the spread of a disturbance across the whole network[24]. As a result, a modular network structure can increase network robustness (e.g., see Liu et al.[25]). Finally, under nestedness, interaction specialists preferentially interact with generalists (i.e., highly connected species)[26]. Nestedness can enhance network robustness against the loss of specialist species if the generalist species at the network core persist[17,18,27,28]. Although the above three network properties are often interrelated, they provide complementary information on network structure and robustness[18,29–31]. Accordingly, we should elucidate the relative impact of habitat fragmentation on each metric and how they interactively influence network robustness.

In fragmented habitats, patch (or habitat island) size and isolation likely affect network structure[14,15,32]. Specialist species are often more sensitive to habitat fragmentation than generalists[33,34] and should be lost first in small and isolated habitats, potentially reducing network nestedness[14,35,36]. Similar expectations can be formulated for network robustness in fragmented landscapes[18,37] because the high diversity of interactions on large islands should lead to lower extinction risks[14,38]. Furthermore, unique fragment environments (e.g., intense competition for food) may influence connectance[36,39], leading to dietary niche expansion and, consequently, adding new links into a network while blurring the boundaries among network modules[36,40]. As a result, we expected that PFNs on small patches have higher connectance, lower modularity and lower nestedness than on large patches, which ultimately leads to lower robustness. Similarly, highly isolated patches are expected to follow a similar pattern as small ones.

Dam construction is becoming a major cause of habitat fragmentation worldwide[41–43]. Hydroelectric dams cause extensive inundation of lowlands, resulting in former hilltops transformed into land-bridge islands of varying areas and isolation. Such reservoir islands surrounded by a homogenous matrix (water) have clear geographic boundaries, the same age and relatively simple biotas[44]. The above features of reservoir islands can circumvent the confounding effects of, e.g., heterogeneous geological times and diverging evolutionary processes[41,45], present in other types of fragmented ecosystems (e.g., forests surrounded by agricultural matrix). Therefore, reservoir islands that have been around for a considerable time represent ideal systems to study the effects of habitat fragmentation on species interaction networks[41,43,46,47]. Despite a few empirical studies that have examined how dam-induced islands affect mutualistic interactions (e.g., ant-plant networks[46]), we still know little about how dam-induced fragmentation affects frugivore-mediated seed dispersal networks.

Here, we assessed the impacts of habitat fragmentation (indicated by island area and isolation) on the diversity, structure and stability of PFNs over two years on 22 land-bridge islands in a large reservoir, the Thousand Island Lake (TIL), in eastern China. The starting forest vegetation in this region was similar among different islands, which was subject to natural secondary succession without human disturbance in the past 60 years (see also Wilson et al.[44]). To overcome the challenges of collecting plant-frugivore interactions simultaneously on separate islands, we used a recently developed technique of arboreal camera trapping, which is a non-invasive, cost-effective method to collect highly resolved data of frugivory interactions at broad spatial scales and fine temporal resolution[47,48]. We address two questions: (1) How do island area and isolation affect species richness, interaction richness, network structure and stability of PFNs? (2) What are the potential mechanisms driving the structure and stability of PFNs in the fragmented forests?

In this study, we used a high-quality dataset of plant-frugivore interactions and assessed the effects of dam-induced habitat fragmentation on the diversity, structure and stability of PFNs. We found that island area explained the changes in assessed metrics consistently better than island isolation from the mainland. Our simulations further suggest that the remaining generalist species (i.e., abundant plants and small-bodied birds) are critical for maintaining community stability on islands, ascribed to their interaction flexibility and the relatively low isolation. In addition, we expanded the concept of interaction release from oceanic islands to reservoir islands. At the same time, our study used a quasi-experimental approach that offers a network perspective for exploring the relationship between biodiversity and ecosystem stability in fragmented landscapes.

## Results
We recorded a total of 10,117 plant-frugivore independent interaction events throughout all study sites (Fig. 1 and Supplementary Table 1), encompassing 402 unique pairwise links among 34 fleshy-fruited plant species and 44 frugivorous bird species (32 omnivores, 11 insectivores and one granivore; Supplementary Tables 2 and 3; Supplementary Data 1). Each network included interactions among 9.04 ± 4.85 (mean ± SD) plant and 14.87 ± 6.54 bird species (Supplementary Table 4).

The surrounding mainland sites had more species of large-bodied fruit-eating birds (i.e., >100 g) than on the study islands (10 vs 8 species). Interaction richness associated with the 10 largest bird species in an aggregated PFN from the mainland sites was ~18% (33/182). On our study islands, ~65% of the interaction richness among the eight largest frugivorous birds occurred on the seven largest islands (>32 ha; Fig. 2a). In contrast, the remaining 15 islands (<10 ha) had only limited large-bodied fruit-eating birds and interaction richness (Fig. 2b).

**Network structure and robustness compared with null models.** The observed values of connectance, modularity and nestedness were significantly different from random networks generated from regional species pools in all networks (Supplementary Table 5 and

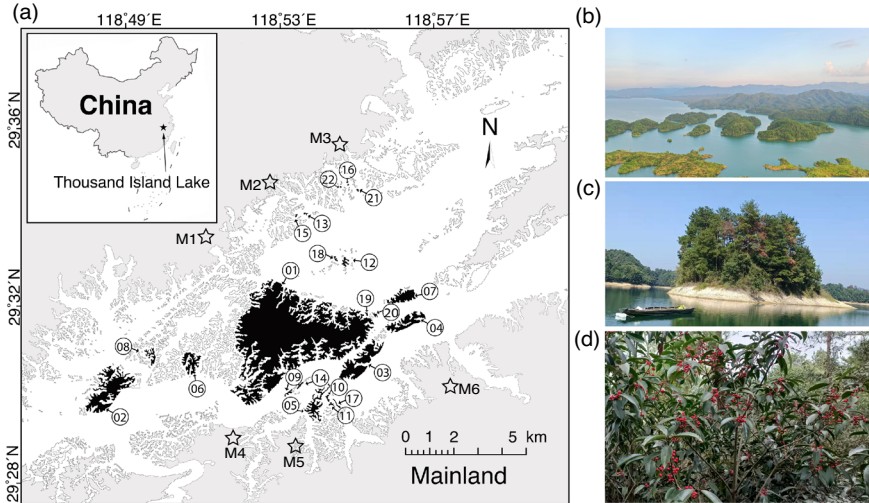

**Fig. 1 Map of the study region and the fragmented landscape of the study system.** The 22 study islands (in black) and six surrounding mainland sites (in pentacle, where 'M' indicates 'Mainland site') in the Thousand Island Lake, Zhejiang, China (the map in panel **a** modified from Si et al.[54]). Islands are labelled by decreasing area, from 01 = largest to 22 = smallest. All unsurveyed islands and the mainland are shown in light grey, with water in white. **b**–**d** show the landscape of the lake, a reservoir island, and a common fruiting plant (*Ilex chinensis*).

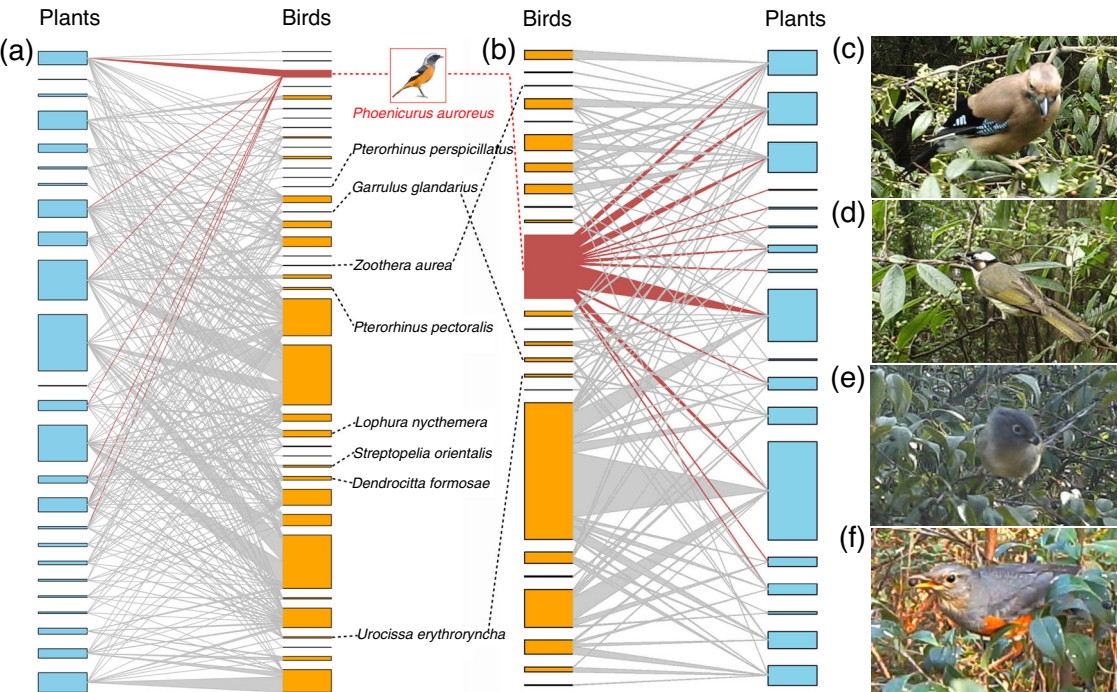

**Fig. 2 Illustration of two aggregated plant-frugivore networks and several representative species in the Thousand Island Lake, China. a** An aggregated network on the seven largest islands (Islands 01–07) and **b** an aggregated network on the seven smallest islands (Islands 16–22). Both networks illustrate the interactions (in grey lines) between avian frugivorous species (in orange) and fleshy-fruited plant species (in light blue). The widths of the grey lines are proportional to interaction frequencies between species. Scientific names of birds between panels **a** and **b** represent eight large-bodied birds higher than 100 g (in black) and one small-bodied bird (in red and illustration with a red border) with high cross-island mobilities. Interactions between *Phoenicurus auroreus* and fruiting plants are marked red in panels **a** and **b**. Representation of several common interacting species in this system: **c** Eurasian Jay (*Garrulus glandarius*) and **d** Light-vented Bulbul (*Pycnonotus sinensis*) feeding on *Vaccinium mandarinorum*. **e** Huet's Fulvetta (*Alcippe hueti*) feeding on *Vaccinium carlesii*. **f** Grey-backed Thrush (*Turdus hortulorum*) feeding on *Rhaphiolepis indica*.

Supplementary Fig. 1). Overall, PFNs were less connected, more modular and less nested than expected by chance from null models, regardless of island attributes. For robustness under the worst-case scenario (i.e., most-abundant plant species are lost first), the robustness of almost all networks on the surveyed islands (except for Islands 13 and 15) significantly differed from null-model expectations. In contrast, nearly half of these networks did not

differ significantly from null models in the following three scenarios: random (plant species are lost randomly), best-case (least-abundant plant species are lost first) and size-case (the largest bird species are lost first; Supplementary Table 5 and Supplementary Fig. 1). Additionally, PFNs had a higher observed mean value of robustness in the size-case scenario than in the random and worst/best-case scenarios (Supplementary Table 5).

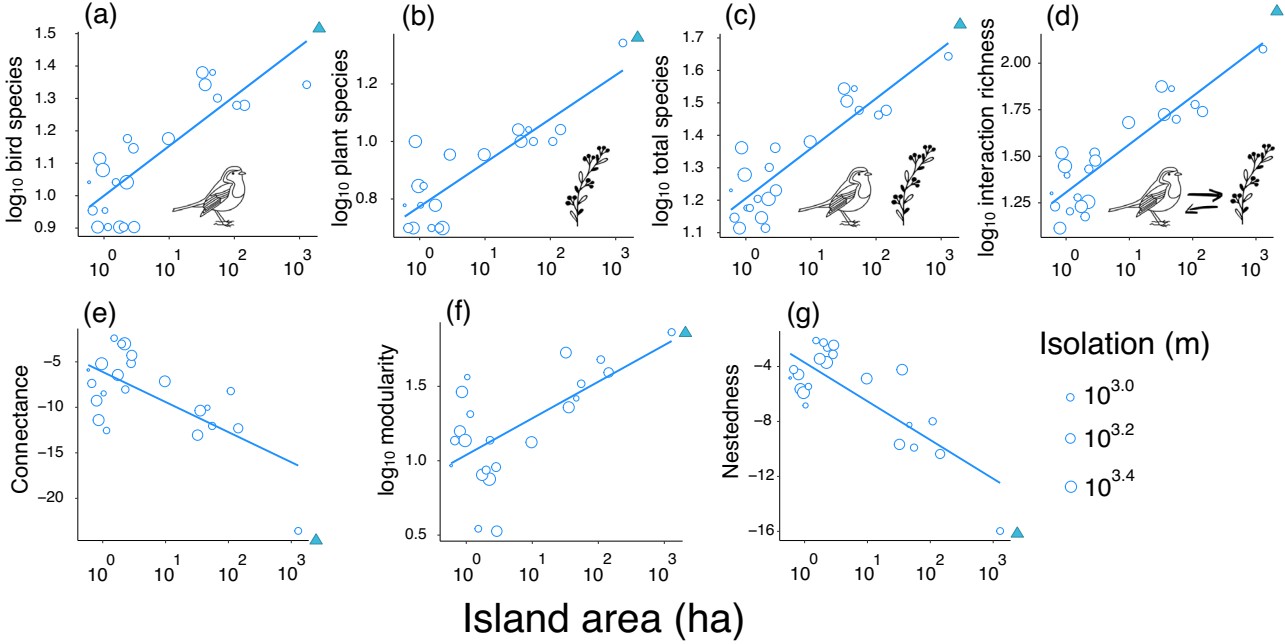

**Fig. 3 The effect of island area and isolation on the diversity (species and interaction) and the structures of plant-frugivore networks in the Thousand Island Lake, China. a–d** show species and interaction richness increasing with island area. **e–g** show three whole-network indices (connectance, modularity and nestedness) changing with island area. Each hollow circle corresponds to a reservoir island, and solid triangles indicate data from the surrounding mainland sites. The lines show the predicted values of diversity and network structural metrics for island areas obtained by holding isolation constant at their means. A greater circle indicates islands with higher isolation. Besides nestedness and connectance, all variables were log₁₀-transformed. Note that network metrics (i.e., connectance, modularity and nestedness) were standardised to z-scores using null models.

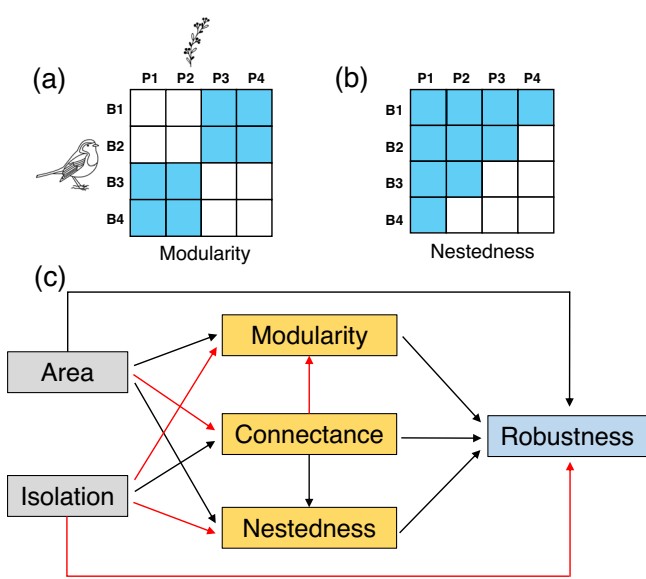

**Fig. 4 Illustration of two network structures and the hypotheses.**
**a** A perfectly modular structure, where modules of species interact more closely with each other, and **b** a perfectly nested structure, where specialist species (e.g., B4 or P4, where 'B' indicates 'Bird', and 'P' indicates 'Plant') interact with a subset of partners of generalist species (e.g., B1 or P1). Blue cells indicate interactions between bird and plant species, while white cells do not. **c** All potential hypothesised effects of habitat fragmentation (island area and isolation) on the structures (connectance, modularity and nestedness) and robustness of plant-frugivore networks. Black and red lines indicate positive and negative effects, respectively.

**Diversity, structure and robustness of PFNs along island area and isolation.** Island area had positive effects on the number of birds, plants, total species, and the number of unique interactions ($p < 0.001$), as well as modularity ($p < 0.01$; Fig. 3 and Supplementary Table 6). However, island area negatively affected network connectance and nestedness ($p < 0.001$). PFNs had the lowest values of connectance and nestedness in surrounding mainland sites, whereas they had the highest diversity (i.e., richness) and modularity (Fig. 3). Both Δ- and z-transformations of network metrics had similar patterns along island area, while the range of Δ-variations was greater on small islands than on large ones (Supplementary Fig. 1). In contrast, island isolation and connectivity did not affect any network metrics (Supplementary Tables 6–12).

We performed structural equation models (SEM) to assess how habitat fragmentation affected the structure and robustness of PFNs (Fig. 4). For the random and worst/size-case scenarios, island area, not isolation or connectivity, affected network structure and ultimately network robustness on study islands (Fig. 5 and Supplementary Tables 13–19). Therefore, while island area did not directly affect robustness, it still indirectly affected network robustness (Supplementary Table 20). Island area was indirectly mediated by modularity and/or connectance under the random scenario (Fig. 5a) and was mediated via connectance in the worst-case scenario (Fig. 5b). In contrast, island area had a positive effect on robustness indirectly mediated by connectance and modularity under the size-case scenario (Fig. 5d). For the best-case scenario, island area had neither a direct nor an indirect effect on robustness (Fig. 5c). Island area both had a weak, negative direct and indirect effect on nestedness, whereas the indirect effect of island area on nestedness was mediated by connectance. Meanwhile, modularity increased with island area,

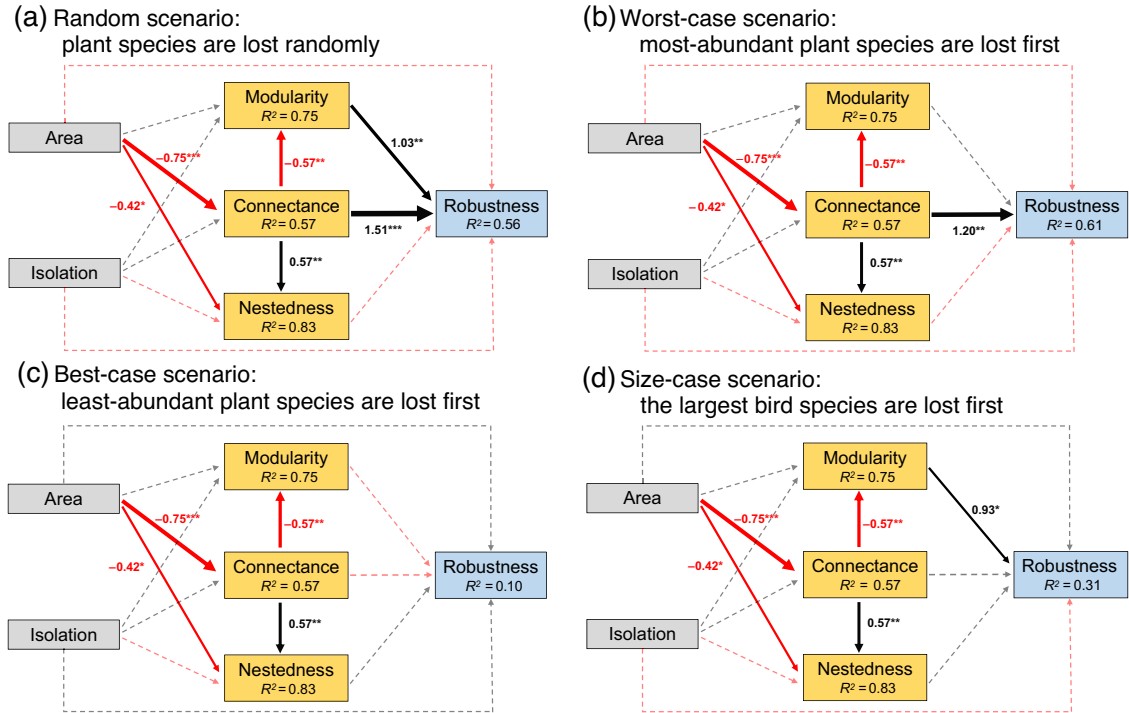

**Fig. 5 Path diagrams for four scenarios of the effects of island area and isolation on the structure and stability of plant-frugivore networks in the Thousand Island Lake, China. a** Random scenario, **b** Worst-case scenario, **c** Best-case scenario and **d** Size-case scenario. The solid lines indicate significant paths, whereas dotted lines represent non-significant paths. Black arrows represent positive effects and red arrows represent negative effects. Numbers alongside each path are standardised coefficients. Significant effects are depicted for *$p < 0.05$, **$p < 0.01$, and ***$p < 0.001$. Conditional $R^2$ values are given in the boxes. Fisher's $C = 2.659$, $p = 0.265$.

again mediated by changes in connectance. Nestedness did not affect network robustness in any of the four extinction scenarios (Fig. 5 and Supplementary Tables 13–20).

## Discussion

This study assessed the effects of island area and isolation on PFNs in fragmented subtropical forests. The results showed that larger islands supported larger networks, with more species and interactions, greater modularity, but lower connectance and nestedness. However, we did not find an effect of island isolation or connectivity on PFNs. Comparing the networks on the largest island and from nearby mainland sites of our study, we found that both the largest island and the mainland sites had similar network structures. However, the aggregated network in the mainland sites had greater species and interaction richness than the networks on the islands, given the approximately similar sampling efforts. Furthermore, PFNs were more vulnerable to three plant extinction scenarios than under a scenario where large-bodied birds were lost. High connectance and/or modularity increased network robustness to plant (i.e., random lost or most-abundant species are lost first) and frugivore extinction (i.e., large-bodied species are lost first). Consequently, island area indirectly determined the robustness of PFNs, mediated by variation in connectance and/or modularity among small and large islands.

Island area positively affected the number of species and interactions. These findings are consistent with species-area and interactions-area relationships[49–51]. Moreover, the interactions-area relationship had a higher slope than the species-area relationship regardless of direct observation or random resampling (Supplementary Tables 6 and 21), which suggests that species interactions are more sensitive than species per se in fragmented landscapes[52]. For area effects, a potential explanation is that

island area can act as a surrogate for space and resource availability (e.g., various habitats and related niches)[14,15,39]. Interestingly, an aggregated PFN from mainland sites had a higher species and interaction richness than that from the largest sampled island in our study, despite the sampling area being similar between both locations (Fig. 3 and Supplementary Table 4). These findings support our hypothesis that habitat fragmentation per se reduced species and interaction richness on islands, beyond the effects of habitat area loss.

Although isolation or connectivity could be essential factors in influencing seed-dispersal networks[32] and plant-herbivore networks[53], we found no effects of isolation in our study in the TIL (Supplementary Tables 6–12). A possible explanation of lacking isolation effects concerns the relatively short inter-island distances (Supplementary Table 22), which may be insufficient to isolate most bird species[54]. For example, two small-bodied winter migratory birds (~15 g), *Tarsiger cyanurus* and *Phoenicurus auroreus*, have been recorded foraging for berries on almost all study islands, indicating the possibility of pervasive movements among islands. Indeed, previous work in TIL also found that island isolation did not affect the plant and breeding bird diversity[44,54]. In addition to inter-island distances, our connectivity assessments indicated that the size of neighbouring islands and plant resource similarities might not be the relevant determinants of PFNs. However, most frugivorous birds in TIL are omnivorous (Supplementary Table 3) and highly mobile, which can explain the lack of isolation effects on PFNs.

Network connectance decreased with increasing island area, as expected. The relationship can be attributed to the greater species richness on larger islands, because connectance typically shows a strong negative correlation with the number of interacting species[18,21,37]. However, our null-model analyses excluded the effect of species richness when estimating connectance. Thus, we

suggest that in addition to the effects of species richness, the negative relationship can further be explained by the prevalence of generalist species (e.g., opportunists) on small islands, i.e., of functionally similar bird species, as indicated by a previous study in the same system[55]. For instance, the Light-vented Bulbul (Pycnonotus sinensis) foraged on all fleshy-fruited plants on small islands (e.g., Islands 12, 14, 18 and 21). Similarly, generalised plants (e.g., Rhaphiolepis indica and Vaccinium carlesii) were consumed by nearly all birds on relatively small islands (e.g., Islands 16 and 17) (Supplementary Data 1). In general, these generalised interacting species on small islands tend to be drought-tolerant plants and small-bodied birds, while some unusual shade-tolerant plants (e.g., Callicarpa giraldii) and relatively large-bodied avian frugivores were rarely recorded (Supplementary Data 1 and Fig. 2). Therefore, rare species might suffer relatively high extinction risks and so-far unpaid extinction debts[42].

In contrast to connectance, network modularity increased with island area, as expected. A plausible reason is that low modularity on small islands was counteracted by high connectance[23,36]. For instance, we found dietary expansion on small islands in TIL (e.g., Daurian Redstart, P. auroreus, fed on almost twice the number of plant species on small islands than on large ones; Fig. 2). We further found that about 75% of widespread frugivorous birds, which were found on more than half of all study islands, increased their normalised degrees with decreasing island area (e.g., the degree-area association for P. auroreus, Pearson's coefficient: $r = -0.71$, p-value < 0.001; Supplementary Table 23), again pointing towards interaction release on small islands. Of course, this expansion trend could be more pronounced on oceanic islands (e.g., the interaction release on the Galápagos Islands), attributed to the scarcity of food resources[36]. In summary, we found strong evidence for interaction release – which was so far only documented for oceanic islands – also on small reservoir islands after dam construction.

Habitat loss (i.e., the decrease of island area) contributed to a nested structure, with networks on smaller islands being more nested than those on larger ones, contradicting our hypothesis. However, similar findings have been reported for seed dispersal networks elsewhere (e.g., in the Atlantic Forest[15]). This greater nestedness may follow from a high connectance (i.e., large dietary overlap between species) and seed dispersal redundancy on small islands[56], as we found that island area had an indirect negative effect on nestedness mediated via connectance, with a comparable effect size of direct (−0.42) and indirect effects (−0.43 [−0.75 × 0.57]; Fig. 5). On the one hand, highly connected species (e.g., P. sinensis) have high contributions to network nestedness[57]. On the other hand, although habitat fragmentation can negatively affect habitat specialists (e.g., some relatively large-bodied birds of Leiothrichidae), low isolation between islands and high bird species turnover on small islands may weaken the negative impact of specialist species loss on nestedness[54].

The effect of network structure on robustness depended on specific extinction scenarios. In this study, network connectance and modularity can individually or jointly promote robustness (Fig. 4a, b, d). When rare plants were lost first, we found that connectance and modularity did not affect robustness (Fig. 5c). Although previous studies found a positive effect between nestedness and robustness[17,26,27,37], we did not detect significant effects in any of the four extinction scenarios (Fig. 5). A potential reason for this finding is that nestedness is sensitive to the thresholds of interaction loss and partners' rewiring ability (e.g., see Grass et al.[17]). Consequently, our simulations provide four different but complementary scenarios to understand the complex relationship between network structure and stability under habitat fragmentation.

When simulating preferential loss of large-bodied birds, we found that PFNs on large islands were more robust than on small islands, mainly mediated through connectance and modularity. A possible explanation is that large-bodied frugivorous birds (e.g., >100 g) and their interactions were relatively rare in the fragmented system, as well as mainly concentrated on large islands (Fig. 2). Moreover, a recent study found that the robustness of PFNs may underestimate the ecological consequences of frugivore extinction[58], so severely fragmented forest patches may be more vulnerable. Consequently, defaunation affected the stability of PFNs in fragmented forests, underscoring the importance of large patches for maintaining forest integrity.

Furthermore, the effect on stability was negligible when simulating the preferential extinction of rare plants, suggesting that rare plants tend to be weak interactors, and that natural systems are relatively resistant to their systematic removal[59]. However, when plants were removed either randomly or by preferential extinction of the most abundant species, we unexpectedly found that PFNs were more robust on small islands than on large ones (Fig. 4a, b). This result may, at first sight, seem surprising. Although network robustness may increase with the proportion of omnivorous birds[14,60–62], we did not find small islands to have more omnivorous birds than large islands in our study (Supplementary Table 4), whereas we found that nine widely distributed frugivorous birds in our study system had greater degrees on small islands (Fig. 2 and Supplementary Table 23). A possible explanation could be the detected pattern of dietary expansion of frugivorous birds on small islands, which may stabilise the remaining network core of closely interacting species.

While our results cannot directly shed light on the provision of seed dispersal services, it is clear that surviving birds on small islands were somewhat resistant to food resource changes (plants removal in random or in order of highest to lowest abundance here)[63,64]. One potential reason is that reservoir islands filter out some vulnerable species[55,65]. As found by Betts et al.[65], environmental changes may eliminate fragmentation-sensitive species that cannot adapt, while surviving species experience selection pressure and adaptive evolution. Consequently, they have a stronger anti-interference ability in their current fragmented habitats. For instance, in our study system, common drought-tolerant plants (e.g., the dominant understory species V. carlesii[54]) and small-bodied generalised birds confer interaction flexibility[40,64], especially on small islands. Indeed, similar findings have been reported from the Kenyan rainforest[66] and other ecosystems[18,67]. Moreover, food resources are concentrated on water-isolated islands, increasing the cost of moving birds between islands to forage, and impelling island-dwelling birds to scatter their dietary preferences to compensate for spatial and temporal variations in food resources[36]. Accordingly, our findings strengthen the predictability of the effects of perturbations on PFNs in disturbed habitats.

Of course, simple metrics of network structure and robustness are insufficient to capture all possible aspects of ecological stability. In general, plant and frugivore communities are thought to be depleted in highly fragmented landscapes, commonly used to justify conservation efforts for forest functionality[2,14,15]. In this study, the random and worst-case scenarios do not mean that small islands are without detrimental effects since they may also have lower functional diversity (e.g., of bird communities[55]). Potentially, habitat fragmentation may accelerate the homogenisation of PFNs, resulting in lower functional resilience[7]. Although it is unlikely that we have exhausted all possible removal scenarios, our study provides crucial insights into understanding the structural stability of PFNs in fragmented landscapes.

## Materials and methods

**Study region.** This study was conducted on subtropical land-bridge islands of the Thousand Island Lake (TIL; 29°22′–29°50′N, 118°34′–119°15′E) in Zhejiang Province, eastern China. The lake (or reservoir) was formed in 1959 after the construction of the Xin'anjiang Dam for hydroelectric production. The lake area is approximately 580 km$^2$, containing 1078 land-bridge islands (previously mountaintops) at the maximum water level (108 m a.s.l.). The dominant vegetation on most islands and the adjacent lowland mainland in TIL is secondary forest, dominated by Masson pine (*Pinus massoniana*), with *Vaccinium carlesii* in the understory[54]. This region has a humid subtropical climate (hot and humid summers, and cool to mild winters), with the rainy season occurring primarily between April and June. The average annual precipitation is ~1,430 mm, and the average annual temperature is 17.0 °C, with daily temperatures ranging from –7.6 in January to 41.8 °C in July[54].

We selected 22 islands in our study region that covered a wide range of island area and isolation (Fig. 1 and Supplementary Table 1). We also selected six nearby mainland sites to compare the structure of island's PFNs that have been simplified due to habitat fragmentation. We constructed an aggregated PFN in the mainland sites, which are continuous habitats not subjected to habitat fragmentation. TIL is dominated by small islands (area <10 ha), and the largest island is Island 01 (area >1000 ha), which we have included (Fig. 1). The selected islands thus represent the natural distribution of island area and isolation in the lake system (see also Si et al.[68]). The area of study islands ranged from 0.59 to 1289.23 ha, and their minimum shore-to-shore distance to the mainland (a widely used variable as isolation in island biogeography) ranged from 640.53 to 3261.96 m. Island area was not correlated to isolation (Pearson's coefficient: $r = –0.15$, $p$-value = 0.51). The inter-island distance ranged from 41 to 11,956 m (Supplementary Table 22). Given the complexity of the measures of isolation in island biogeography, we additionally considered six fragment connectivity parameters (more details in Santos et al.[53] and Si et al.[54]) to quantify island connectivity (*sensu* isolation) (Supplementary Table 24), while isolation (the distance to the mainland here) did not affect connectivity (Supplementary Fig. 2). In this study, the specific connectivity metrics combine geographical distances among islands (Supplementary Table 22), neighbouring island area (Supplementary Table 1), and similarity in plant resource composition among islands (Supplementary Table 25) that was calculated by the R package 'vegan' v2.5-7.

We used the proportional sampling method[69] to set up transects on study islands to sample plant-frugivore interactions. Specifically, eight transects were set on the largest island with an area >1000 ha, four transects on islands between 100 and 1000 ha, two transects on islands between 10 and 100 ha, and one on each of the remaining islands with an area <10 ha (c. 1 ha for most islands; Supplementary Table 1). Each transect had a width of 20 m and length of 200–400 m (Supplementary Table 1), covering various fleshy-fruited plants in different forest strata. The transects run along the island ridge from the edge to the interior[47,48]. Besides the island transects, we set up a total of 11 transects in surrounding mainland sites, keeping the sampling area approximately equal to the area of the largest island (Island 01).

**Sampling plant-frugivore interactions.** The main fruiting season of our study region is from July to next January[47]. Thus, we sampled plant-frugivore interactions along transects on each island over two complete fruiting seasons (i.e., from July 2019 to January 2020, and from July 2020 to January 2021) using the recently developed technique of arboreal camera trapping (see more details in Zhu et al.[47,48]). Arboreal camera trapping can record the plant-frugivore interactions simultaneously and continuously on a set of islands at broad scales and with fine temporal resolutions that are not attainable with traditional manual observations, e.g., using binoculars[48]. We monitored fleshy-fruited plant species that depend mainly on seed dispersers because birds in this study are the dominant taxa dispersing seeds among islands (see also Liu et al.[70]). We systematically searched for fleshy-fruited plants along each transect, tagged them with ID cards, and recorded the coordinates of each plant individual. We then monitored all tagged fruiting plant individuals by installing the remote camera traps when the fruits began to mature. In this process, we did not include some individuals of plant species with rare fruits because fruits were too scarce to yield meaningful data on interactions (more details in Zhu et al.[48]).

We installed 588 effective infrared digital cameras (LTL 6210MC, Lieke Company, China) at 0.5–8 m above the ground (Supplementary Table 1). The specific locations of cameras depended on the height of fruiting branches. We aimed the cameras toward target branches with high fruit densities to maximise the detection probabilities of visiting frugivores[48,71]. Cameras were separated by at least 20 m to reduce oversampling[47]. We configured cameras to work continuously, 24 h a day, and record three photos followed by a 10-second video when triggered, with a 10-s delay between triggers to alleviate the rapid consumption of memory cards when animals remained in the camera field of view. The cameras can automatically stamp the date and time on photos and videos. We checked the camera's battery life and memory card capacity every two weeks (see more details in Zhu et al.[47]). We manually retrieved species identities and the number of frugivorous bird individuals, reviewing photos and videos. We classified the foraging behaviours of avian frugivores and considered fruit swallowing and pecking as valid frugivory events[72], which is a conservative way to identify avian

frugivores (see also Fig. 2 in Zhu et al.[47]). Thus, we identified them as bird-fruit interactions. Meanwhile, we defined an independent interaction event as consecutive photos/videos of the same plant-frugivore interaction, separated by more than five minutes[73], sensu Si et al.[74]. Accordingly, the interaction frequency was calculated as the number of independent frugivory events recorded during the entire sampling period[75].

**Sampling completeness and sampling effect.** We organised the data into a matrix to construct interaction networks for each island. Each row represented a plant species, each column represented a frugivorous bird, and cell values represented interaction frequency[75]. We used individual-based extrapolation/interpolation methods to estimate sampling coverage across islands to evaluate sampling completeness[76]. We treated 'abundance' as the interaction frequencies recorded for each pairwise link[40]. We performed this analysis using the *iNEXT* function in the R package 'iNEXT' v2.0.20[77]. The sampling completeness was high, ranging from 81.7% to 82.4% on all islands, indicating that our sampling of interactions was sufficient on each island and that there was no bias in sampling completeness with changing island area (Supplementary Fig. 3).

We further used a null modelling approach to test whether the sampling effect (i.e., different camera days on each island) potentially affected the species richness and interaction richness across islands. Overall, the total sampling effort was 4889 camera days in six nearby mainland sites and 25,676 camera days on study islands, whereby Island 22 had the smallest number of camera days, i.e., 414 camera days (Supplementary Table 1). We thus randomly subsampled the data on islands with >414 camera days 1000 times and constructed the PFNs based on the resampled data using rarefaction analyses. We calculated the mean number of birds, plants, total species and interaction richness derived from 1000 resampled networks for each island. We found that the relationships between rarefied diversity metrics and island variables were consistent with the observed patterns (Supplementary Tables 6 and 21), indicating that sampling effects did not bias our analyses. All following analyses were therefore done using the original data.

**Network metrics.** To explore changes in network structure and robustness with island area and isolation, we calculated a set of complementary network-level metrics for each of the 22 interaction networks. Specifically, we quantified richness (bird species, plant species and unique interactions), network structures (connectance, modularity, and nestedness) and stability (robustness). Among them, (i) connectance describes the density of interactions, which was calculated as the proportion of observed unique pairwise interactions relative to all possible interactions (i.e., the product of the number of plants and birds) in a network[21]; (ii) Modularity was calculated using the DIRTLPAwb+ algorithm[78]; (iii) Nestedness was computed as the weighted nestedness (wNODF) based on the overlap of interactions and decreasing fill[79]; and (iv) robustness was calculated as the area below the secondary extinction curve, from simulated co-extinctions between plant species and their frugivore partners[28]. We assumed that there are bottom-up (plant-mediated) effects[12,17] in PFNs, as justified by Borrvall et al.[61], Scherber et al.[80] and Schleuning et al.[81]. Moreover, a global analysis also found that about 50% of extinct plants occur on islands, but the extinctions are species-specific[82]. Hence, we first simulated plant species' extinctions and measured co-extinctions of frugivores, and then simulated a directed animal extinction induced by defaunation (i.e., size-case scenario: prioritised extinctions of large-bodied bird species[83]), as justified by Donoso et al.[58], Rumeu et al.[63] and Rogers et al.[84]. Specifically, plant species extinctions were assigned randomly until the total collapse of a given interaction network. The extinction simulations were repeated 1000 times for each network, and robustness values were averaged across simulations. In addition, given that the more-abundant plant species surviving in the remaining patches may be drought-tolerant and generalised, rare species tend to be habitat-specific or relatively vulnerable[70], we thus sequentially removed plants according to two potential extinction orders: a worst-case scenario with species' extinctions ordered from highest to lowest abundance, and a best-case scenario with a reversed order. We pooled plant abundance data from surveyed transects on each island to obtain plant removal orders (Supplementary Table 25). We used the *second.extinct* function for extinction simulations, and all network metrics were calculated in the R package 'bipartite' v2.16[85].

To exclude the effect of different network sizes (i.e., the total number of interacting species) of various islands on metrics of network structure and robustness calculated above, we used Patefield's null modelling method[86]. This method reshuffles interactions among species but keeps the total number of bird and plant species and their interaction sums in each network on an island unchanged[86]. We ran the null model 1,000 times by using the *nullmodel* function in the R package 'bipartite' and calculated null model corrections, including Δ- and z-transformations (z-scores) of each network metric (Supplementary Table 5), which both represent the difference between observed and null-model obtained values[60]. Specifically, all z-scores were calculated by subtracting the mean value of 1000 randomisations from the observed value (Obs) and dividing the result by the standard deviation (Sd) of expected values (Exp): $\Delta_{mean}/Sd_{null}$, where $\Delta_{mean} = Obs – Exp_{null(1…n)}$. These z-scores can directly compare network metrics across islands, excluding potentially confounding effects of differences in network size[15,31], while Δ can quantify the extent an empirical observation departs from a pattern expected by a specific null model. Furthermore, we also used Patefield's null model to

evaluate whether network structural metrics and robustness differed from a random distribution of interactions[72]. A given metric was considered significantly non-random if the observed value did not overlap with the confidence interval (95%).

**Statistics and reproducibility**. We used multiple linear regression models to assess the effects of island area and isolation (accounting for connectivity) on the richness (species and their interactions), structures (connectance, modularity and nestedness) and stability of PFNs. We then examined the relationships of z-scores (network structural metrics and robustness) and the direct/indirect effects of island area and isolation on z-scores using piecewise structural equation modelling[87] based on linear regression models (see Fig. 4c). This approach allows elucidating causal relationships between variables to be constructed and tested, even with relatively low sample sizes[17,87]. Pathways were summarised to understand which structural attributes determine network robustness (Fig. 4). SEMs were implemented in the R package 'piecewiseSEM' v2.1.2[87]. All the remaining variables, except connectance, nestedness and robustness, were log$_{10}$-transformed before the analyses to normalise the model fit[88]. All variables used in the SEMs were scaled and centred. All statistical analyses were conducted in R software v4.0.5[89].

**Reporting summary**. Further information on research design is available in the Nature Portfolio Reporting Summary linked to this article.

## Data availability
Data for analysing plant-frugivore interactions are provided in the supplementary data.

## Code availability
All R packages and software versions needed to reproduce the results are presented in the Methods.

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

## Acknowledgements

We thank Shupei Tang, Pen Han, Guangpeng Wei, Tong Lin, Xue Zhang, Xinyu Xu, Zhengguang Fan, Jingjing Wang, Junjie Shao, Chang Cai, Qi Si and other members in the Si Lab of East China Normal University and the Ding Lab of Zhejiang University for their assistance in the field and in processing photos and videos. We also thank Xin'an River Ecological Development Group Corporation, Chun'an Forestry Bureau and the Thousand Island Lake National Forest Park for permits to conduct this research. This research was supported by grants from the National Natural Science Foundation of China (#32071545, #32030066 and #31872210), the Program for Professor of Special Appointment (Eastern Scholar) (#TP2020016), Baishanzu National Park Scientific Research Program (#2021ZDLY03) and the Shanghai Rising-Star Program (#19QA1403300).

## Author contributions

X.S. designed the study. W.L. and X.S. conceived the ideas. W.L., C.Z., D.W. and Y.K. conducted the fieldwork. W.L. analysed data with assistance from Y.Z. and D.Z.; W.L. led the writing with contributions from I.G., D.P.V., P.D. and X.S. All authors contributed to the revisions and comments, and approved the publication.

## Competing interests

The authors declare no competing interests.
