## [Peer Review File · Communications Biology]

Reviewers' comments:

Reviewer #1 (Remarks to the Author):

COMMSBIO-22-1627-T

Title: Island area indirectly reduces the robustness of plant–frugivore networks through connectance in fragmented forests

Comments for authors

GENERAL COMMENTS

This is a very-well written article assessing the effect of island area and isolation on diversity, structure and robustness of plant-frugivore networks constructed through focal observations gathered with camera traps. The study has been carried out on the Thousand Island Lake (TIL); a system of land-bridge remnant islands artificially formed in 1959 after the construction of a Dam. I enjoyed reading this manuscript, and I do not have serious concerns. The paper addresses an issue relevant to the fields of Network Ecology, Landscape Ecology, Island Biogeography and Conservation Biology, and should be of interest to the readers of Communications Biology. However, I have some questions whose answers I hope can be reflected in the manuscript to improve it. In addition, the Discussion section is 8.5 pages long. I strongly recommend to make a synthesis effort in this section, which also becomes a bit confusing in some parts (see detailed comments below).

DETAILED COMMENTS

INTRODUCTION:

L 43: typo in “represents”, replace by “represent”

L 56-75 (paragraph focused on network properties): From this paragraph it is not clear that in Network Ecology, the Robustness of a network can be calculated (as the area below the extinction curve generated by secondary extinctions). In other words, it seems that Robustness is something that we can infer from the three network properties that will be calculated (Connectance, Modularity and Nestedness). Thus, I suggest to replace in L59 “aspect of assessing” by “metric to assess”.

L 95-97: I agree with this sentence. But it is important to highlight that considerable time should be elapsed between the anthropogenic and sudden formation of these islands and the assessment of the fragmentation effects (the biota should “adjust” to the new situation). Otherwise, you could be assessing plant-frugivore networks that are the result of a previous situation. See the concept of extinction debt (e.g., Hanski and Ovaskainen 2001, Valiente-Banuet et al. 2015, doi: 10.1111/1365-2435.12356).

L 101-103. It would be useful here to know whether the starting vegetation formation is the same in the whole area (besides being explained in the “Study region” section L.367-369).

MATERIAL AND METHODS:

L 369-370: rewrite

L 373-376: According to Supplementary Table 1, it seems that the distribution of island areas is skewed towards small islands. Only one out 22 islands is larger than 1000 ha, the second category ranges from 100-200 ha (n = 2), and the rest of islands (n = 19) are < 100 ha. Fourteen out of these latter are < 10 ha. These categories were used to establish the number of transects. I suggest briefly justifying the reason for this unbalance.

L 375: replace “way” by “variable”

L 378: To be consistent, give the range of the inter-island distance.

L 381 Supplementary Table 7: Improve the Foot note of this table: "follows the methodology proposed by...)

L 395: I think that "of an island" can be removed.

L 420: replace "are" by "were"

L 422-425: I don't understand why the pecking of fruits are considered as seed dispersal events. There are bird species that are fruit peckers and they do not disperse seeds. In fact, I do not understand why you mention seed dispersal, since what it is being recorded are frugivory events.

L440 Supplementary Fig. 3: Why are there five panels in this figure? And what is the rationale behind grouping the islands in that way?

L464-467: According to previous research based on simulation models, mutualistic networks are more sensitive to plant extinctions but, what is more probable in your study system? I mean, from the discussion it is inferred that large-bodied frugivores like hornbills and toucans (L.197), or Leiiothrichidae bird species (L216-217) face a high risk to extinction. Is that risk higher than the risk of plant extinctions? In other words, I think that the simulation of species' extinctions should have a better justification, it must have a biological basis.

L 481: "including" instead of "include"?

L 488: "can quantify to what extent"

L 496: "accounting for" instead of "included"? Add a parenthesis after "structure" to include Connectance, Modularity and Nestedness.

RESULTS

My only comment is that I suggest to keep the order of Connectance, Modularity and Nestedness constant. These metrics appear in this order in the introduction, and it would be helpful to report the results (also in Fig. 2) and discuss such results in the same order whenever possible.

DISCUSSION

As previously said in the Overall comments, the discussion is too long. It is important to make a synthesis effort.

L. 151: our results showed THAT larger...

L 153: lower network connectance and nestedness.

L185: "foraging" instead of "that foraged"

L 188: I suggest to cite here the specific studies about plant and breeding bird communities (e.g. Si et al. 2014 J. Biogeogr. 41; for birds). On the other hand, check in the reference list: Wilson et al. 2016 (not 2015).

L 192: "from a fragmented" instead of "from the fragmented"

L195-198: There is some confusing information regarding connectivity and extinctions. For instance, L 181-188: distance among islands is relatively short, so some birds have the potential for moving over the water surface (and connect islands). But here it is said that there is a lack of the role of connectivity due to the lost of hornbills or toucans that can serve as island connectors. On the other hand, is there previous information on the extinction dynamics of species in your study system? (This comment is in line with my previous comment on the extinction simulation analysis).

L. 198: serving as 'connectors'.

L229: Cite Fig.4 I can't see how do you obtain the value -0.43.

L240-249: I am really curious about what could be the pattern in the nearest continental area. It would be great to compare your results with a continuous landscape with more or less the same plant-frugivore community. I am not asking you to do so, but it would be very interesting.

L 279-282. Cite those previous studies. Consider the inclusion of Rumeu et al. 2017 *Funct. Ecol.*, doi: 10.1111/1365-2435.12897.

L 296-297: I do not understand this sentence.

L334-337: Do you know how was the composition of the initial fleshy-fruited community? It would be interesting to analyze how the fleshy fruited community has changed after the formation of the dam.

Fig. 2, L733: ...and one small-body sized bird (Daurian Redstart, *Phoenicurus aureus*) with high island mobility (marked by red grid). The red boxes and their links indicate the fruiting plants it feeds on.

Fig. 3 Refer to Supplementary Table 10 in the Figure caption.

Fig. 4 Add in the figure legend the meaning of each scenario (e.g. B, Worst-case scenario: most-abundant species are lost first).

Congrats for your contribution.
Best regards,
Beatriz Rumeu

Reviewer #2 (Remarks to the Author):

The paper by Li and colleagues presents plant-frugivore network data from a set of islands that resulted from dam creation in China. The observations enable the authors to test a pressing question for ecology and conservation – how does fragmentation affect networks of ecological interactions? The observations enable a quasi-experimental way to address this question, which very few studies have been able to achieve in the context of species interactions or interaction networks. Overall I found the paper to be effective, including its writing and figures, and feel the paper will make an important contribution to the field. My one concern involves the overall interpretation highlighted in the title. This interpretation made is that more severe fragmentation makes ecological networks more robust, according to coextinction simulations developed using the observational data. The implication of this particular result--that fragmentation is beneficial for ecological networks--runs contrary to the broader literature on habitat fragmentation and, based on my reading, is not consistent with the observational data. Interpreting the observed data only, it appears that that greater fragmentation substantially reduces species and interaction richness and comes at the detriment of species with specialized interactions and large frugivores. I feel that a simpler interpretation, that emphasizes the observational data rather than the simulation model results on robustness, would simplify and strengthen the paper and better play to the advantages of this strong observational dataset and study design.

To explain my one larger comment in greater detail, this focuses on the key interpretation that's highlighted in the title of the paper and around line 36-38 in the abstract. My first issue is that the wording may not allow the interpretation to come across clearly for readers. Perhaps the wording is a little ambiguous because the result is unexpected or difficult to interpret. Two ways to interpret this result are that 1) fragmentation creates more robust networks or 2) following fragmentation, what's left is a set of generalist species that maintain a highly connected (and nested) core of plant-frugivore interactions. This second interpretation is made starting around line 323, but is emphasized much less than the first interpretation.

While I understand that coextinction simulations offer a way to understand the behavior of mutualistic networks as they undergo disruption, I feel that a key advantage of the study system is that it is able to directly observe how mutualistic networks respond to real-world disruption (in this case, a spectrum of habitat fragmentation severity). To me, focusing on robustness estimated

through simulations emphasizes the simulation models to the detriment of the real-world observations on the response of networks to disruption. An interpretation that is more ecologically straight-forward to me, in line with other expectations from ecology and conservation biology, and focused on observations rather than simulation results would make the title of the paper something more like "Plant-frugivore network degradation under habitat fragmentation leaves a small core of interacting generalists" or "Plant-frugivore network degradation under habitat fragmentation reduces interaction diversity, including with specialized interactors and large-bodied frugivores." So overall, I feel that some tweaks to the interpretation of results to emphasize the empirical results would strengthen the paper.

Reviewers' comments:

Reviewer #1:

GENERAL COMMENTS

This is a very-well written article assessing the effect of island area and isolation on diversity, structure and robustness of plant–frugivore networks constructed through focal observations gathered with camera traps. The study has been carried out on the Thousand Island Lake (TIL); a system of land-bridge remnant islands artificially formed in 1959 after the construction of a Dam. I enjoyed reading this manuscript, and I do not have serious concerns. The paper addresses an issue relevant to the fields of Network Ecology, Landscape Ecology, Island Biogeography and Conservation Biology, and should be of interest to the readers of Communications Biology. However, I have some questions whose answers I hope can be reflected in the manuscript to improve it. In addition, the Discussion section is 8.5 pages long. I strongly recommend to make a synthesis effort in this section, which also becomes a bit confusing in some parts (see detailed comments below).

Response: We thank the reviewer for these encouraging words. In this version, we have shortened the Discussion, and rephrased the confusing sentences to improve the clarity in writing.

INTRODUCTION:

L 43: typo in “represents”, replace by “represent”

Response: Done.

L 56-75 (paragraph focused on network properties): From this paragraph it is not clear that in Network Ecology, the Robustness of a network can be calculated (as the area below the extinction curve generated by secondary extinctions). In other words, it seems that Robustness is something that we can infer from the three network properties that will be calculated (Connectance, Modularity and Nestedness). Thus, I suggest to replace in L59 “aspect of assessing” by “metric to assess”.

Response: Thanks for catching this point! We have replaced the wording “aspect of

assessing” by “metric to assess” as suggested in Line 55.

L 95-97: I agree with this sentence. But it is important to highlight that considerable time should be elapsed between the anthropogenic and sudden formation of these islands and the assessment of the fragmentation effects (the biota should “adjust” to the new situation). Otherwise, you could be assessing plant-frugivore networks that are the result of a previous situation. See the concept of extinction debt (e.g., Hanski and Ovaskainen 2001, Valiente-Banuet et al. 2015, doi: 10.1111/1365-2435.12356).

Response: Good suggestions! We have clarified these sentences in Lines 89–90 as:

‘... reservoir islands, which have been around for a considerable time (the biota should ‘adjust’ to new situations), represent ideal systems to study the effects of habitat fragmentation on species interaction networks.’

L 101-103. It would be useful here to know whether the starting vegetation formation is the same in the whole area (besides being explained in the “Study region” section L.367-369).

Response: The primary forests in this region were selectively or clearly cut with organized logging in the 1950s, resulting in almost complete deforestation before the lake inundation. After logging, airplanes sowed native pines, and the vegetation is naturally secondary succession without human disturbance. Importantly, the region has been protected since 1962 (Wilson et al. 2016), thus, the starting vegetation should be similar in all islands in this region.

In this version, we included a sentence in Lines 97–99 mentioning that the forest vegetation was similar throughout our study region:

‘The starting forest vegetation in this region was similar among different islands, which was subject natural secondary succession without human disturbance in the past 60 years (see also⁴⁴)’

MATERIAL AND METHODS:

L 369-370: rewrite

Response: We have clarified this sentence in Lines 296–297 as:

‘This region has a humid subtropical climate (hot and humid summers, and cool to mild winters), with the rainy season occurring primarily between April and June.’

L 373-376: According to Supplementary Table 1, it seems that the distribution of island areas is skewed towards small islands. Only one out of 22 islands is larger than 1000 ha, the second category ranges from 100-200 ha (n = 2), and the rest of islands (n = 19) are < 100 ha. Fourteen out of these latter are < 10 ha. These categories were used to establish the number of transects. I suggest briefly justifying the reason for this unbalance.

Response: We selected the study islands based on the natural distribution of island areas in this lake system, which is dominated by small islands. We added a sentence in Lines 311–314 to justify the reason for this unbalance:

‘TIL is dominated by small islands (area < 10 ha), and the largest island is Island 01 (area > 1000 ha), which we have included (Fig. 5). The selected islands thus represent the natural distribution of island area and isolation in the lake system (see also⁶⁹).’

L 375: replace “way” by “variable”

Response: Done.

L 378: To be consistent, give the range of the inter-island distance.

Response: Amended.

L 381 Supplementary Table 7: Improve the Foot note of this table: “follows the methodology proposed by...)

Response: Done.

L 395: I think that “of an island” can be removed.

Response: Done.

L 420: replace “are” by “were”

Response: Done.

L 422-425: I don't understand why the pecking of fruits are considered as seed dispersal events. There are bird species that are fruit peckers and they do not disperse seeds. In fact, I do not understand why you mention seed dispersal, since what it is being recorded are frugivory events.

Response: Sorry for the confusing writing. We have replaced “seed dispersal events” by “frugivory events” (see Line 351).

L440 Supplementary Fig. 3: Why are there five panels in this figure? And what is the rationale behind grouping the islands in that way?

Response: We initially visualized the curves in five panels in order to minimize overlapping among curves. However, in the new version of Fig. S3, we merged the previous five panels together, and also added sampling completeness curves of mainland sites into Fig. S3.

L464-467: According to previous research based on simulation models, mutualistic networks are more sensitive to plant extinctions but, what is more probable in your study system? I mean, from the discussion it is inferred that large-bodied frugivores like hornbills and toucans (L.197), or Leiotherichidae bird species (L216-217) face a high risk to extinction. Is that risk higher than the risk of plant extinctions? In other words, I think that the simulation of species' extinctions should have a better justification, it must have a biological basis.

Response: Thanks for these useful suggestions! We have supplemented and improved the Methods, Results, and Discussion in the simulation section, especially to improve the consistency of extinction simulation scenarios between Methods and Discussion.

We summarized these revisions in three points:

1) we have justified three scenarios of simulated plant loss in the Methods, and we

added a possible extinction scenario, which is that bird species are extinct from the largest- to smallest-body sized, as suggested in Lines 389–395 and Lines 398–400:

Lines 415–421: *'We assumed that there are bottom-up (plant-mediated) effects^{12,17} in PFNs, as justified by Borrvall et al.⁶⁶, Scherber et al.⁸⁰, and Schleuning et al.⁸¹. Moreover, a global analysis also found that about 50% of extinct plants occur on islands, but the extinctions are species-specific⁸². Hence, we first simulated plant species' extinctions and measured co-extinctions of frugivores, and then simulated a directed animal extinction induced by defaunation (i.e., size-case scenario: prioritized extinction of large-bodied bird species⁸³), as justified by Donoso et al.⁵⁸, Rumeu et al.⁶⁰, and Rogers et al.⁸⁴.'*

L424–426: *'..., given that the more-abundant plant species surviving in the remaining patches may be drought-tolerant and generalized, rare species tend to be habitat-specific or relatively vulnerable⁷⁰, ...'*

2) We compared the simulated bird extinction scenario with three plant extinction scenarios. We found that network robustness was higher in the bird extinction scenario than in three plant extinction scenarios, indicating that plant-frugivore networks were more sensitive to plant extinction than birds. We thus added the following sentences in Lines 125–127 and Lines 155–156:

Lines 125–127: *'Additionally, PFNs had a higher observed mean value of robustness in the size-case scenario than in the random and worst/best-case scenarios (Supplementary Table 8).'*

L155–156: *'Furthermore, PFNs were more vulnerable to three plant extinction scenarios than a scenario where large-bodied birds are lost.'*

3) We supplemented and rephrased the Discussion with some potentially vulnerable plant species in fragmented landscapes, as suggested in Lines 196–200:

*'In general, these generalized interacting species tend to be drought-tolerant plants and small-bodied birds on small islands, while some uncommon shade-tolerant plants (e.g., *Callicarpa giraldii*) and relatively large-bodied avian frugivores were rarely recorded*

on small islands (Supplementary Data 1; Fig. 2), so that rare species might suffer relatively high extinction risks, as they may represent unpaid extinction debts (i.e., further species extinctions⁴²).'

L 481: “including” instead of “include”?

Response: Done.

L 488: “can quantify to what extent”

Response: Done.

L 496: “accounting for” instead of “included”? Add a parenthesis after “structure” to include Connectance, Modularity and Nestedness.

Response: Done.

RESULTS

My only comment is that I suggest to keep the order of Connectance, Modularity and Nestedness constant. These metrics appear in this order in the introduction, and it would be helpful to report the results (also in Fig. 2) and discuss such results in the same order whenever possible.

Response: Agreed. We used the same order of ‘connectance, modularity and nestedness’ throughout the manuscript.

DISCUSSION

As previously said in the Overall comments, the discussion is too long. It is important to make a synthesis effort.

Response: In this version, we have re-worked the Discussion for a better flow, and cut 796 words to shorten it.

L. 151: our results showed THAT larger...

Response: Thanks! We added the word “that” in Line 152. We also carefully checked

the whole manuscript to correct this grammatical error.

L 153: lower network connectance and nestedness.

Response: Done.

L185: “foraging” instead of “that foraged”

Response: Done.

L 188: I suggest to cite here the specific studies about plant and breeding bird communities (e.g. Si et al. 2014 J. Biogeogr. 41; for birds). On the other hand, check in the reference list: Wilson et al. 2016 (not 2015).

Response: The related references were cited and corrected (see Line 176).

L 192: “from a fragmented” instead of “from the fragmented”

Response: Done.

L195-198: There is some confusing information regarding connectivity and extinctions. For instance, L 181-188: distance among islands is relatively short, so some birds have the potential for moving over the water surface (and connect islands). But here it is said that there is a lack of the role of connectivity due to the lost of hornbills or toucans that can serve as island connectors.

On the other hand, is there previous information on the extinction dynamics of species in your study system? (This comment is in line with my previous comment on the extinction simulation analysis).

Response: Sorry for the confusion regarding the wording of ‘connectivity’. In our manuscript, short distances among islands mean high connectivity, which is the geographical connectivity of isolated islands in this fragmented landscape. The connectors in ecological networks represent the role of species, not islands in our system. In this version, we rephrased the descriptions of these two meanings, and used ‘species connectors’ specifically in the sentences describing network structures (see

Line 184).

Our study system has previous information on local extinction dynamics (e.g., a species occupied an island in a year but was absent in the next year) based on our annual bird survey datasets. However, we found that small islands have fewer large-bodied bird species (e.g., > 100 g) than large islands (or surrounding mainland sites). Therefore, we also considered the scenario of preferential extinction of large-bodied birds in this revised version as the reviewer suggested (see the results in the Panel d of Fig. 4). We believe that this additional analysis can provide a complementary understanding of network robustness in our study system.

L. 198: serving as ‘connectors’.

Response: Done.

L229: Cite Fig.4 I can't see how do you obtain the value -0.43.

Response: This value of -0.43 was calculated by multiplying the coefficients of two paths (island area \rightarrow connectance, -0.75 , and connectance \rightarrow nestedness, 0.57). We revised this sentence in Lines 217–218 as:

‘... with a comparable effect size of direct (-0.42) and indirect effects (-0.43 [-0.75×0.57]; Fig. 4)’

L240-249: I am really curious about what could be the pattern in the nearest continental area. It would be great to compare your results with a continuous landscape with more or less the same plant-frugivore community. I am not asking you to do so, but it would be very interesting.

Response: Good suggestion! We have sampled frugivory interactions in the surrounding mainland sites, although we didn't report the results of mainland sites in the previous version. The sampling area in the mainland sites is approximately equal to our chosen largest study island (Island 01).

In this version, we followed the reviewer's suggestion and added an aggregated plant-frugivore network from six surrounding mainland sites for comparison. Network

information in the mainland sites (e.g., species, fruit-eating interactions, structure, and robustness) has been included to the main text (Lines 112–113 and Lines 132–133) and the Supplementary Information (Lines 39–40, Lines 67–68, Lines 78–79, Lines 83–84, Lines 89–90, and Lines 102–103). We also included associated analyses, such as the sampling completeness in Supplementary Figure 3 and multiple regressions in Fig. 3 (triangles in the panels) for mainland sites in Line 48 and Line 681.

Lines 112–113: *‘The surrounding mainland sites had more species of large-bodied fruit-eating birds (e.g., > 100 g) than on the study islands.’*

L132–133: *‘PFNs had the lowest values of connectance and nestedness in surrounding mainland sites, whereas had the highest diversity (i.e., richness) and modularity (Fig. 3).’*

L 279-282. Cite those previous studies. Consider the inclusion of Rumeu et al. 2017 *Funct. Ecol*, doi: 10.1111/1365-2435.12897.

Response: The suggested reference was cited. See Lines 253 & 395.

L 296-297: I do not understand this sentence.

Response: This sentence was deleted.

L334-337: Do you know how was the composition of the initial fleshy-fruited community? It would be interesting to analyze how the fleshy fruited community has changed after the formation of the dam.

Response: Agreed, but unfortunately, we don’t have this dataset — China was experiencing a special period in the 1960s, and many scientific data were not collected at that time. However, as we replied to the comment L101–103, we can confirm that the background of the vegetation formation was similar in this region.

Fig. 2, L733: ...and one small-body sized bird (Daurian Redstart, *Phoenicurus auroreus*) with high island mobility (marked by red grid). The red boxes and their links indicate the fruiting plants it feeds on.

Response: Corrected. See Line 675.

Fig. 3 Refer to Supplementary Table 10 in the Figure caption.

Response: Done.

Fig. 4 Add in the figure legend the meaning of each scenario (e.g. B, Worst-case scenario: most-abundant species are lost first).

Response: Done. We added a simulation on the loss of avian frugivores (Fig. 4d) as suggested (see Line 691).

Congrats for your contribution.

Best regards,

Beatriz Rumeu

Response: We thank the reviewer again for the constructive comments and suggestions.

Reviewer #2:

The paper by Li and colleagues presents plant-frugivore network data from a set of islands that resulted from dam creation in China. The observations enable the authors to test a pressing question for ecology and conservation – how does fragmentation affect networks of ecological interactions? The observations enable a quasi-experimental way to address this question, which very few studies have been able to achieve in the context of species interactions or interaction networks. Overall I found the paper to be effective, including its writing and figures, and feel the paper will make an important contribution to the field.

Response: We thank the reviewer for their positive comments.

My one concern involves the overall interpretation highlighted in the title. This interpretation made is that more severe fragmentation makes ecological networks more robust, according to coextinction simulations developed using the observational data.

The implication of this particular result--that fragmentation is beneficial for ecological networks—runs contrary to the broader literature on habitat fragmentation and, based on my reading, is not consistent with the observational data. Interpreting the observed data only, it appears that that greater fragmentation substantially reduces species and interaction richness and comes at the detriment of species with specialized interactions and large frugivores. I feel that a simpler interpretation, that emphasizes the observational data rather than the simulation model results on robustness, would simplify and strengthen the paper and better play to the advantages of this strong observational dataset and study design.

Response: Agreed. In this version, we interpreted the results more carefully and based on our strong observational data. Accordingly, we retitled our manuscript to “*Plant-frugivore network degradation under habitat fragmentation leaves a small core of interacting generalists*”.

To explain my one larger comment in greater detail, this focuses on the key interpretation that’s highlighted in the title of the paper and around line 36-38 in the abstract. My first issue is that the wording may not allow the interpretation to come across clearly for readers. Perhaps the wording is a little ambiguous because the result is unexpected or difficult to interpret. Two ways to interpret this result are that 1) fragmentation creates more robust networks or 2) following fragmentation, what’s left is a set of generalist species that maintain a highly connected (and nested) core of plant-frugivore interactions. This second interpretation is made starting around line 323, but is emphasized much less than the first interpretation.

Response: Thanks again for the comments. We have clarified the wording by mainly following the second way of the interpretation as suggested. In this version, for the first interpretation, we have removed this confusing sentence in original Lines 312–314: ‘*This finding implied that increased levels of anthropogenic disturbance (habitat loss here) could lead to increased tolerance of seed dispersal networks to future perturbations.*’). We also rephrased the wording regarding the network robustness that

occurred on small islands during dynamic simulations. For example, the original sentence ‘... PFNs on fragmented islands with relatively small area can exhibit higher robustness to random or worse-case plant extinctions.’ was replaced by ‘... surviving birds on small islands had somewhat resistant to changes in food resources (plants removal in random or in order of highest to lowest abundance here)^{60,61}.’ in Lines 251–253. We believe this change will make it easier to understand.

To strengthen the second interpretation, we cited a very relevant article (Betts et al. 2019. Science), and we added two sentences in Lines 257–262: ‘As found by Betts et al.⁶⁴, environmental changes may have eliminated fragmentation-sensitive species that could not adapt, while surviving species experienced selection pressure and adaptive evolution. Consequently, they have a stronger anti-interference ability in their current fragmented habitats. Similarly, in our study system, common drought-tolerant plants (e.g., a dominant understory species *V. carlesii*⁵⁴) and small-bodied generalized birds confer interaction flexibility^{40,61}, especially on small islands.’

While I understand that coextinction simulations offer a way to understand the behavior of mutualistic networks as they undergo disruption, I feel that a key advantage of the study system is that it is able to directly observe how mutualistic networks respond to real-world disruption (in this case, a spectrum of habitat fragmentation severity). To me, focusing on robustness estimated through simulations emphasizes the simulation models to the detriment of the real-world observations on the response of networks to disruption. An interpretation that is more ecologically straight-forward to me, in line with other expectations from ecology and conservation biology, and focused on observations rather than simulation results would make the title of the paper something more like “Plant-frugivore network degradation under habitat fragmentation leaves a small core of interacting generalists” or “Plant-frugivore network degradation under habitat fragmentation reduces interaction diversity, including with specialized interactors and large-bodied frugivores.” So overall, I feel that some tweaks to the interpretation of results to emphasize the empirical results would strengthen the paper.

Response: Agreed. As we replied in previous comments, we now focused more on the observational results in this version following the suggestions from Reviewer #2. First, we changed the title to '*Plant-frugivore network degradation under habitat fragmentation leaves a small core of interacting generalists*'. Second, for the Abstract, we have included the loss of specialized interactors and large-bodied frugivores under habitat fragmentation. We also have emphasized that habitat fragmentation results in remnant species forming generalized networks, and pointed out that conservation of large forest remnants has an important role in the support of interaction diversity and forest functionality (see Lines 32–38). Third, we have largely revised the Discussion: (i) We have shortened the length of the Discussion, reducing the number of words from 2564 to 1768. (ii) We have explained the properties of generalized species in fragmented landscapes as well as some species with a relatively high risk of extinction (see Lines 196–200). (iii) We have highlighted the positions occupied by the observed large-bodied bird species in plant-frugivore networks (see Lines 238–244), and we have increased this interpretation of extinction filters under habitat fragmentation (see Lines 257–262). We believe that our manuscript was substantially improved by these adjustments.

We list these specific adjustments amended in the main manuscript as below:

Lines 32–38: '*We found that the degradation of plant-frugivore networks under habitat fragmentation reduces interactions, including those with specialized interactors and large-bodied frugivores. Habitat fragmentation results in remnant species forming generalized networks centred on small-bodied birds and abundant plants, especially on small islands, as well as more connected, less modular, and more nested networks, and interaction release. Our results reveal the importance of preserving large forest remnants to support more interaction diversity and forest functionality.*'

L196–200: '*In general, these generalized interacting species tend to be drought-tolerant plants and small-bodied birds on small islands, while some uncommon shade-tolerant plants (e.g., *Callicarpa giraldii*) and relatively large-bodied avian frugivores were rarely recorded on small islands (Supplementary Data 1; Fig. 2), so that rare*

species might suffer relatively high extinction risks, as they may represent unpaid extinction debts (i.e., further species extinctions⁴²).

L238–244: *‘A possible explanation is that large-bodied frugivorous birds (e.g., >100 g) and their interactions were relatively rare in the fragmented system, as well as these species were mainly concentrated on large islands (Fig. 2). Moreover, a recent study found that the robustness of PFNs may underestimate the ecological consequences of frugivore extinction⁵⁸, so severely fragmented forest patches may be more vulnerable. Consequently, defaunation affected the stability of PFNs in fragmented forests and underscored the importance of large patches for maintaining forest integrity.’*

L257–262: *‘As found by Betts et al.⁶⁴, environmental changes may have eliminated fragmentation-sensitive species that could not adapt, while surviving species experienced selection pressure and adaptive evolution. Consequently, they have a stronger anti-interference ability in their current fragmented habitats. Similarity, in our study system, common drought-tolerant plants (e.g., a dominant understory species *V. carlesii*⁵⁴) and small-bodied generalized birds confer interaction flexibility^{40,61}, especially on small islands.’*

References

12. Pocock, M. J. O., Evans, D. M. & Memmott, J. The robustness and restoration of a network of ecological networks. *Science* **335**, 973–977 (2012).
17. Grass, I., Jauker, B., Steffan-Dewenter, I., Tschamtko, T. & Jauker, F. Past and potential future effects of habitat fragmentation on structure and stability of plant-pollinator and host-parasitoid networks. *Nat. Ecol. Evol.* **2**, 1408–1417 (2018).
40. Vizentin-Bugoni, J. et al. Structure, spatial dynamics, and stability of novel seed dispersal mutualistic networks in Hawai'i. *Science* **364**, 78–82 (2019).
42. Jones, I. L., Bunnefeld, N., Jump, A. S., Peres, C. A. & Dent, D. H. Extinction debt on reservoir land-bridge islands. *Biol. Conserv.* **199**, 75–83 (2016).

44. Wilson, M. C. et al. Habitat fragmentation and biodiversity conservation: key findings and future challenges. *Landscape Ecol.* **31**, 219–227 (2016).
54. Si, X., Pimm, S. L., Russell, G. J., Ding, P. & Burns, K. C. Turnover of breeding bird communities on islands in an inundated lake. *J. Biogeogr.* **41**, 2283–2292 (2014).
58. Donoso, I. et al. Downsizing of animal communities triggers stronger functional than structural decay in seed-dispersal networks. *Nat. Commun.* **11**, 1582 (2020).
60. Rumeu, B. et al. Predicting the consequences of disperser extinction: richness matters the most when abundance is low. *Funct. Ecol.* **31**, 1910–1920 (2017).
61. Wong, B. B. M. & Candolin, U. Behavioral responses to changing environments. *Behav. Ecol.* **26**, 665–673 (2015).
64. Betts, M. G. et al. Extinction filters mediate the global effects of habitat fragmentation on animals. *Science* **366**, 1236–1239 (2019).
66. Borrvall, C., Ebenman, B. & Jonsson, T. Biodiversity lessens the risk of cascading extinction in model food webs. *Ecol. Lett.* **3**, 131–136 (2000).
69. Si, X. et al. The importance of accounting for imperfect detection when estimating functional and phylogenetic community structure. *Ecology* **99**, 2103–2112 (2018).
70. Liu, J. et al. The distribution of plants and seed dispersers in response to habitat fragmentation in an artificial island archipelago. *J. Biogeogr.* **46**, 1152–1162 (2019).
80. Scherber, C. et al. Bottom-up effects of plant diversity on multitrophic interactions in a biodiversity experiment. *Nature* **468**, 553–556 (2010).
81. Schleuning, M. et al. Ecological networks are more sensitive to plant than to animal extinction under climate change. *Nat. Commun.* **7**, 13965 (2016).
82. Humphreys, A. M., Govaerts, R., Ficinski, S. Z., Nic Lughadha, E. & Vorontsova, M. S. Global dataset shows geography and life form predict modern plant extinction and rediscovery. *Nat. Ecol. Evol.* **3**, 1043–1047 (2019).
83. Dirzo, P. et al. Defaunation in the Anthropocene. *Science* **345**, 401–406 (2014).
84. Rogers, H. S., Donoso, I., Traveset, A. & Fricke, E. C. Cascading impacts of seed disperser loss on plant communities and ecosystems. *Annu. Rev. Ecol. Evol. Syst.*

52, 641–666 (2021).

Reviewers' comments:

Reviewer #1 (Remarks to the Author):

COMMSBIO-22-1627A

Title: Plant-frugivore network degradation under habitat fragmentation leaves a small core of interacting generalists

Comments for authors

This is the second time I read this manuscript, and I am glad to see that my previous suggestions helped to improve this version. Authors have addressed all my concerns and suggestions. Besides, I agree with comments made by reviewer #2, which highlight the suitability of the study system to emphasize the results from observational data.

Yet, I still have some minor comments:

Title: I agree this is a better title than before. Maybe replace "degradation" by "simplification".

L 34-36. The last part of this sentence is confusing. Perhaps include a stop after "small islands" and rewrite the remaining information. I recommend a detailed revision of the English throughout the manuscript.

L 88 Replace ", which" by "that"

L 298-306. This paragraph is out of place. It should go after the next paragraph ("We selected...").

L 307 (onwards). As I suggested in the previous version, I think that including mainland sites is very interesting, especially because it allows a comparison of the islands' PFNs (which have been simplified due to fragmentation) with a continuous "control" PFN (not subjected to fragmentation). However, the motivation behind including these mainland sites should be expressed in this paragraph. Accordingly, I would delete the insertion you made in line 307 ("six nearby mainland sites and") and would add a sentence after: Besides, we also selected six nearby mainland sites in order to...

On the other hand, the incorporation of the mainland sites to the manuscript deserves some discussion (to add in the "Discussion" section of the ms). For instance, are the patterns found in mainland similar to those found on the largest island of the TIL system?

L 323 Sampling plant-frugivore interactions. How frequently did you check for the battery life of the cameras or whether memory cards were full?

L 365 report also the sampling effort in the mainland sites.

L 110-112 What about interaction richness with large-sized frugivores on mainland sites? Replace e.g. within the brackets by i.e.

L 120 It would not be amiss to remind the reader what is the worst-case scenario. Idem in L. 122 (random and best/size scenarios).

L 193-198. Please, split this long sentence in two in order to ease the reading.

L 203-204 It is very interesting that you found dietary expansion on small islands in TIL. You mention two bird species, but is this a general trend among frugivorous species shared between large and small islands? Did you detect an increase in the degree of frugivorous birds with the reduction of island size?

L 246-247 A possible explanation for the increase of robustness on small island could be the pattern of dietary expansion detected.

L 262-265 In relation with my previous comment, although small islands do not have more omnivorous bird species than large, perhaps these species have a greater degree in small islands.

Fig. 1. Caption of this figure contains redundant information.

Fig. 2. Caption of this figure should include letters (a) and (b)

Best wishes,
Beatriz Rumeu

Reviewer #2 (Remarks to the Author):

The authors have carefully revised the manuscript and I feel have addressed the issues brought up in the reviews.

Reviewers' comments:

Reviewer #1 (Remarks to the Author):

COMMSBIO-22-1627B

Title: Plant-frugivore network degradation under habitat fragmentation leaves a small core of interacting generalists

Comments for authors

This is the second time I read this manuscript, and I am glad to see that my previous suggestions helped to improve this version. Authors have addressed all my concerns and suggestions. Besides, I agree with comments made by reviewer #2, which highlight the suitability of the study system to emphasize the results from observational data.

Response: Thanks very much for the appraisal.

Yet, I still have some minor comments:

Title: I agree this is a better title than before. Maybe replace “degradation” by “simplification”.

Response: Done. We retitled it as: *‘Plant-frugivore network simplification under habitat fragmentation leaves a small core of interacting generalists’*.

L 34-36. The last part of this sentence is confusing. Perhaps include a stop after “small islands” and rewrite the remaining information. I recommend a detailed revision of the English throughout the manuscript.

Response: Agreed. We have clarified these sentences in Lines 34–36 as:

‘Small islands had more connected, less modular, and more nested networks that consisted mainly of small-bodied birds and abundant plants, as well as showed evidence of interaction release (i.e., dietary expansion of frugivores).’

As suggested, we have carefully checked the text and made a thorough revision of the language.

L 88 Replace “, which” by “that”

Response: Done.

L 298-306. This paragraph is out of place. It should go after the next paragraph (“We selected...”).

Response: Agreed. We have moved this paragraph after the next one. See Lines 327–335.

L 307 (onwards). As I suggested in the previous version, I think that including mainland sites is very interesting, especially because it allows a comparison of the islands’ PFNs (which have been simplified due to fragmentation) with a continuous “control” PFN (not subjected to fragmentation). However, the motivation behind including these mainland sites should be expressed in this paragraph. Accordingly, I would delete the insertion you made in line 307 (“six nearby mainland sites and”) and would add a sentence after: Besides, we also selected six nearby mainland sites in order to...

Response: Thanks again for this good suggestion! We have explained why mainland sites are selected in Lines 309–313:

‘We selected 22 islands in our study region that covered a wide range of island area and isolation (Fig. 5; Supplementary Table 1). We also selected six nearby mainland sites to compare the structure of island’s PFNs that have been simplified due to habitat fragmentation. We constructed an aggregated PFN in the mainland sites, which are continuous habitats not subjected to habitat fragmentation.’

On the other hand, the incorporation of the mainland sites to the manuscript deserves some discussion (to add in the “Discussion” section of the ms). For instance, are the patterns found in mainland similar to those found on the largest island of the TIL system?

Response: We added two sentences in Lines 157–161 and Lines 174–178 to discuss the network structure from the mainland sites:

L157–161: *‘Comparing the networks on the largest island and from nearby mainland sites of our study, we found that both the largest island and the mainland sites had similar network structures. However, the aggregated network in the mainland sites had*

greater species and interaction richness than the networks on the islands, given the approximately similar sampling efforts.'

L174–178: *'Interestingly, an aggregated PFN from mainland sites had a higher species and interaction richness than that from the largest sampled island in our study, despite the sampling area being similar between both locations (Fig. 3; Supplementary Table 4). These findings support our hypothesis that habitat fragmentation per se reduced species and interaction richness on islands, beyond the effects of habitat area loss.'*

L 323 Sampling plant-frugivore interactions. How frequently did you check for the battery life of the cameras or whether memory cards were full?

Response: We checked the battery life and memory card every two weeks. We include this information in Lines 359–360. A more detailed introduction about the protocol of sampling plant-frugivore interactions is available in Ref. 47.

'We checked the camera's battery life and memory card capacity every two weeks (see more details in Zhu et al.⁴⁷).'

L 365 report also the sampling effort in the mainland sites.

Response: Done. See Lines 381–383:

'Overall, the total sampling effort was 4,889 camera days in six nearby mainland sites and ... (Supplementary Table 1).'

L 110-112 What about interaction richness with large-sized frugivores on mainland sites?

Replace e.g. within the brackets by i.e.

Response: Done. See the revisions in Lines 111–113:

'The surrounding mainland sites had more species of large-bodied fruit-eating birds (i.e., > 100 g) than on the study islands (ten vs eight species). Interaction richness associated with the ten largest bird species in an aggregated PFN from the mainland sites was about 18% (33/182).'

L 120 It would not be amiss to remind the reader what is the worst-case scenario. Idem

in L. 122 (random and best/size scenarios).

Response: We have made corresponding supplementary explanations for several simulation scenarios, as suggested in Line 122 and Lines 125–126.

L122: ‘... *worst-case scenario (i.e., most-abundant plant species lost first), ...*’

L125–126: ‘... *in the following three scenarios: random (plant species are lost randomly), best-case (least-abundant plant species are lost first), and size-case (the largest bird species are lost first) ...*’

L 193-198. Please, split this long sentence in two in order to ease the reading.

Response: We have split this long sentence into two sentences (see Lines 202–206).

‘In general, these generalized interacting species on small islands tend to be drought-tolerant plants and small-bodied birds, while some unusual shade-tolerant plants (e.g., Callicarpa giraldii) and relatively large-bodied avian frugivores were rarely recorded (Supplementary Data 1; Fig. 2). Therefore, rare species might suffer relatively high extinction risks and so-far unpaid extinction debts⁴².’

L 203-204 It is very interesting that you found dietary expansion on small islands in TIL. You mention two bird species, but is this a general trend among frugivorous species shared between large and small islands? Did you detect an increase in the degree of frugivorous birds with the reduction of island size?

Response: Thanks for these useful suggestions! In this version, we performed further correlation analysis on the degree of twelve frugivorous birds (e.g., Daurian Redstart, *Phoenicurus aureus*, and Orange-flanked Bluetail, *Tarsiger cyanurus*) and island area, and we found that the degree of nine fruit-eating birds increased as the reduction of island area. Therefore, we added the correlation results between the normalised degree and island area to Supplementary Table 25 (see Supplemental Material), while we added a sentence in Lines 211–214.

L211–214: ‘*We further found that about 75% of widespread frugivorous birds, which were found on more than half of all study islands, increased their normalised degrees with decreasing island area (e.g., the degree-area association for P. aureus, Pearson’s*

coefficient: $r = -0.71$, $p\text{-value} < 0.001$; Supplementary Table 25), again pointing towards interaction release on small islands.'

L216–218: *'In summary, we found strong evidence for interaction release – which was so far only documented for oceanic islands – also on small reservoir islands after dam construction.'*

L 246-247 A possible explanation for the increase of robustness on small island could be the pattern of dietary expansion detected.

Response: Agreed. We revised the sentence in Lines 259–261:

'A possible explanation could be the detected pattern of dietary expansion of frugivorous birds on small islands, which may stabilize the remaining network core of closely interacting species.'

L 262-265 In relation with my previous comment, although small islands do not have more omnivorous bird species than large, perhaps these species have a greater degree in small islands.

Response: Agreed. We added a sentence in Lines 255–259.

'Although network robustness may increase with the proportion of omnivorous birds^{14,60–62}, we did not find small islands to have more omnivorous birds than large islands in our study (Supplementary Table 4), whereas we found that nine widely distributed frugivorous birds in our study system had greater degrees on small islands (Fig. 2; Supplementary Table 25).'

Fig. 1. Caption of this figure contains redundant information.

Response: In this version, we have removed redundant information in the caption. See Lines 669–676:

*'**Fig. 1 Illustration of two network structures and the hypotheses.** Top: Two bipartite networks: (a) a perfectly modular network, where modules of species interact more closely with each other, and (b) a perfectly nested network, where specialist species (e.g., B4 or P4, where 'B' indicates 'Bird', and 'P' indicates 'Plant') interact with a*

subset of partners of generalist species (e.g., B1 or P1). Blue cells indicate interactions between bird and plant species, while white cells do not; Bottom: (c) All potential hypothesized effects of habitat fragmentation (island area and isolation) on the structure (connectance, modularity, and nestedness) and robustness of plant-frugivore networks. Black and red lines indicate positive and negative effects, respectively.'

Fig. 2. Caption of this figure should include letters (a) and (b)

Response: Done. See Lines 680–684:

'(a) An aggregated network on the seven largest islands (Islands 01–07) and (b) an aggregated network on the seven smallest islands (Islands 16–22). Both networks illustrate the interactions (in grey lines) between avian frugivorous species (in orange) and fleshy-fruited plant species (in light blue). The widths of the grey lines are proportional to interaction frequencies between species.'

Best wishes,

Beatriz Rumeu

Response: We thank the reviewer again for the constructive comments and suggestions.

Reviewer #2 (Remarks to the Author):

The authors have carefully revised the manuscript and I feel have addressed the issues brought up in the reviews.

Response: Thanks!

References

14. de Assis Bomfim, J., Guimarães Jr., P. R., Peres, C. A., Carvalho, G. & Cazetta, E.

Local extinctions of obligate frugivores and patch size reduction disrupt the structure of seed dispersal networks. *Ecography* **41**, 1899–1909 (2018).

47. Zhu, C. et al. Arboreal camera trapping: a reliable tool to monitor plant-frugivore

- interactions in the trees on large scales. *Remote Sens. Ecol. Conserv.* **8**, 92–104 (2022).
60. Dalsgaard, B. et al. Opposed latitudinal patterns of network-derived and dietary specialization in avian plant-frugivore interaction systems. *Ecography* **40**, 1395–1401 (2017).
61. Borrvall, C., Ebenman, B. & Jonsson, T. Biodiversity lessens the risk of cascading extinction in model food webs. *Ecol. Lett.* **3**, 131–136 (2000).
62. Liao, J. et al. Robustness of metacommunities with omnivory to habitat destruction: disentangling patch fragmentation from patch loss. *Ecology* **98**, 1631–1639 (2017).